# Carbon Ions for Hypoxic Tumors: Are We Making the Most of Them?

**DOI:** 10.3390/cancers15184494

**Published:** 2023-09-09

**Authors:** Olga Sokol, Marco Durante

**Affiliations:** 1Biophysics Department, GSI Helmholtzzentrum für Schwerionenforchung, Planckstraße 1, 64291 Darmstadt, Germany; o.sokol@gsi.de; 2Institute for Condensed Matter Physics, Technische Universität Darmstadt, Hochschulstraße 8, 64289 Darmstadt, Germany

**Keywords:** hypoxia, carbon ions, CIRT, LET painting, particle therapy, radiotherapy

## Abstract

**Simple Summary:**

Carbon-ion radiotherapy is a potential elective treatment option for hypoxic tumors. Its high linear energy transfer enables enhanced cell killing in radiation-resistant tumors, while the Bragg peak ensures precise targeting. Clinical evidence in pancreatic and cervical cancers supports positive outcomes of carbon treatments. However, the power of carbon ions against tumor hypoxia is generally underexploited and should be considered to improve the clinical benefit.

**Abstract:**

Hypoxia, which is associated with abnormal vessel growth, is a characteristic feature of many solid tumors that increases their metastatic potential and resistance to radiotherapy. Carbon-ion radiation therapy, either alone or in combination with other treatments, is one of the most promising treatments for hypoxic tumors because the oxygen enhancement ratio decreases with increasing particle LET. Nevertheless, current clinical practice does not yet fully benefit from the use of carbon ions to tackle hypoxia. Here, we provide an overview of the existing experimental and clinical evidence supporting the efficacy of C-ion radiotherapy in overcoming hypoxia-induced radioresistance, followed by a discussion of the strategies proposed to enhance it, including different approaches to maximize LET in the tumors.

## 1. Introduction

In the field of cancer treatment, carbon-ion beams are usually considered an attractive option for cases in which conventional radiotherapy approaches have limitations, offering improved tumor control while preserving patients’ quality of life. According to the statistics of the Particle Therapy Cooperation Group [1], there are 14 centers operating in Europe and Asia that provide cancer treatments with accelerated carbon ions (C-ions). Furthermore, after the first patient treatments with various heavy ions at Lawrence Berkeley National Laboratory (LBNL) in 1975–1992 [2], carbon therapy will soon be back in the USA, with a facility about to open at the Mayo Clinic in Jacksonville [3].

Carbon ions, like protons, exhibit a characteristic dose distribution known as the Bragg peak, depositing most of their energy at the tumor site, thereby minimizing damage to critical organs beyond the target area [4]. When compared to light ions, they have a smaller lateral dose penumbra at greater depths, which makes treatment plans more conformal than in proton therapy [5]. However, the main benefit of C-ion radiotherapy (CIRT) lies in its superior biological effectiveness, which is attributed to the densely ionizing nature of C ions in the Bragg peak region [6]. Ionization density is described by linear energy transfer (LET), which is proportional to z^2^/β^2^ [4] and is therefore especially high for heavy, slow ions. High-LET radiation induces a higher fraction of direct DNA damage compared to X-rays, where most of the damage is caused by free radicals produced in water, and the lesions are more difficult to repair [6]. From the treatment planning point of view, this enhanced cell kill leads to an increase in the peak-to-plateau ratio in the spread-out Bragg peak (SOBP, a region of a uniform dose in the target volume) and allows for an increase in biological tumor dose without causing additional normal tissue toxicity. Among the drawbacks of C ions is the dose ‘tail’ in the healthy tissue beyond the SOBP that is generated by the projectile fragments having a similar velocity and direction but a longer range [4]. Furthermore, the benefits of the sharp SOBP edges and high RBE might be jeopardized by the beam-range uncertainties or target motion, compromising the target dose conformity or increasing normal tissue toxicities [4,7].

In recent years, C-ion physics research has tended toward investigations addressing treatment precision, speed, and cost [8], while the field of radiobiology actively studies the impact of C-ion irradiation on the tumor microenvironment and signaling pathways [7,9,10,11]. 

One of the main reasons to use ions heavier than protons in the original Berkeley pilot study was the possibility of overcoming hypoxia-induced radioresistance [12]. In the Berkeley trial, ions such as heavy as argon were used, but toxicity was too high. Carbon was selected first in Japan and then in Europe because it has relatively low LET in the entrance channel and relatively high LET in the SOBP [6]. However, in current clinical practice, the mean LET in the target region is often too low to induce a significant sensitization of hypoxic tumors. In this contribution, we explore potential strategies to optimize CIRT treatment outcomes for hypoxic tumors, seeking to enhance its efficacy.

## 2. Tumor Hypoxia and Radioresistance

Hypoxia [13] is a characteristic feature of solid tumors associated with their disrupted and heterogeneous vascular network, which is known to correlate with poor prognosis in cancer patients. Briefly, tumors develop their vascular system via angiogenesis, utilizing the blood supply from the host organ. However, in tumors, there is no appropriate balance between pro- and antiangiogenic signals [14], and the tumor neovasculature is rather chaotic, dilated, and leaky, sharing both the regular and chaotic features of venules, arterioles, and capillaries [15]. Thus, it can no longer meet the metabolic demands of the developing tumor, leading to the formation of oxygen-deficient regions [16]. There is no strict and universal hypoxia threshold for all tumor types; however, at partial oxygen pressure (pO_2_) levels below approximately 35 mmHg, physiological activities and functions have been suggested to become progressively restricted [17]. 

Lack of oxygen can have a significant impact on the production of circulating tumor cells (CTCs) and on the metastatic capacity of cancer cells [18]. Hypoxia-induced epithelial–mesenchymal transition contributes to CTC generation, promoting migratory and invasive properties [19,20]. Furthermore, poorly oxygenated regions in the primary tumor induce angiogenesis-related gene expression, facilitating CTC entry into the bloodstream [21,22]. 

Hypoxia triggers the activation of hypoxia-inducible factors (HIFs), including HIF-1, which regulates the expression of dozens of genes and mediates pathways influencing metabolism, angiogenesis, cell growth and differentiation, survival, and apoptosis [23,24]. HIFs are found to be elevated in various cancer types [25], making them important targets for pharmacological intervention [26]. 

Two types of hypoxia are usually distinguished: chronic and acute. Chronic, or diffusion-limited hypoxia, caused by limitations in oxygen diffusion from tumor microvessels was first suggested based on histological data from bronchus carcinoma patients [27]. On the other hand, acute (perfusion-limited) hypoxia is caused by temporary complete or partial blood vessel shutdown, leading to fluctuations in microvascular oxygen supply [28]. Solid tumors often contain regions of intermittent or cycling hypoxia with spatial and temporal fluctuations in oxygen levels [29,30,31].

Regardless of its type (although there is evidence of acute hypoxia having a stronger impact [32]), hypoxia affects radiotherapy outcomes due to the crucial role of oxygen in the biological effectiveness of radiation. Already in the 1950s, it was demonstrated that tissues with increased oxygenation are more radiosensitive than hypoxic tissues [33]. This effect is usually explained by the oxygen fixation hypothesis [34,35], which postulates that most DNA can be repaired after being damaged by radicals produced during the interaction of radiation with biological matter; however, that repair is more difficult or impossible (becomes ‘fixed’) when caused by the product of a radical and an oxygen molecule (peroxyl radical). 

In radiotherapy, hypoxia-induced radioresistance is typically described by the oxygen enhancement ratio (OER), i.e., the ratio between the radiation dose in hypoxia and the radiation dose in fully oxygenated conditions (air) resulting in the same biological effect. In conventional radiotherapy, OER can reach a maximum value of 3 [36]. The decrease in OER observed with increasing levels of pO_2_ in tissue is characterized by a sigmoid curve [37].

## 3. CIRT for Hypoxic Tumors: Evidence of Effectiveness 

### 3.1. Decrease in OER with Increasing Particle LET: Mechanisms, In Vitro Data, and In Silico Studies

Multiple approaches have been proposed to mitigate hypoxia-induced radioresistance in recent decades, including approaches involving agents that either increase oxygen delivery or radiosensitize or preferentially kill the hypoxic cells, physics-based approaches such as dose painting by boosting the radiation dose to the hypoxic areas, and the use of high-LET heavy ions. As previously noted, heavy ions produce damage predominantly through direct interaction with the biological targets and are therefore less dependent on free radical production and surrounding oxygen concentrations [38]. In addition, at high LET, there is an increased production of molecular oxygen capable of reacting with DNA and damaging it further after the passage of an ion, i.e., locally causing an oxygenated radiation response [39,40]. This explanation is commonly referred to as the “oxygen in the track” hypothesis. In fact, free radical recombination occurs frequently along tracks with high ionization density [41]. In addition, the O_2_ track concentrations calculated by Monte Carlo techniques compare very well with the O_2_ concentrations estimated from the “effective” amounts of oxygen needed to produce the observed reduction in OER [42], supporting the “oxygen in the track” hypothesis [39]. 

The track effects and production of reactive oxygen species modulate HIF expression [43,44], which is reduced following C-ion irradiation compared to X-ray irradiation [45,46]. Carbon ions seem to be able to induce equal cell killing in chronic and acute hypoxia, while in anoxia, the cells are more resistant in acute conditions [47,48]. 

Experiments with high-LET neutrons and alpha particles showed a decrease in OER compared to X-rays already many years ago. A full OER–LET relationship was originally measured at the LBNL [49] and later confirmed at NIRS (now QST) in Japan [50]. The measured OER values ranged from 3 (photons) to 1 for heavy ions at LET > 300 keV/μm in cells irradiated with ^12^C or ^20^Ne ions. A later modeling study [51] utilized a large pool of experimental OER values measured for particles ranging from protons to argon ions in the LET range of less than 1 keV/µm up to almost 1 MeV/µm. Experimental works in recent years have further expanded the in vitro datasets, including the OER dependency on both particle LET and cell oxygenation [52].

At present, there are multiple models based mainly on in vitro data that attempt to describe the OER(LET, pO_2_) dependencies [51,52,53,54,55,56,57] and include them in the optimization of treatment plans (see the example in Figure 1). All models predict an approximately twofold decrease in the OER starting from LET ~100 keV/µm, dropping down to 1 at values > 200–300 keV/µm.

### 3.2. Carbon Radiation for Hypoxic Tumors: Preclinical Studies

Pioneering studies conducted at the LBNL were the first to measure OER after exposure to heavy ions [58,59] in vivo. The authors compared the efficacy of Si, Ne, and C ions in the treatment of rat rhabdomyosarcoma tumors, showing that OER was reduced with increasing beam LET in animals in hypoxia, which was induced by nitrogen gassing of the animal.

A few decades later, a study employing a xenograft model of human non-small cell lung cancer further demonstrated substantial differences between X-ray and C-ion radiation; the latter induced a ninefold reduction in HIF-1α levels and significantly delayed tumor growth [60]. Studies in Japan [61,62] included mice with squamous cell carcinoma cells injected into the hind limbs, with hypoxia induced by limb clamping. In vivo–in vitro colony assays of the extracted irradiated tumor cells further validated the reduced OER values of samples irradiated with carbon in comparison to those exposed to X-rays. An interesting observation pertained to the dynamics of OER decrease: in vitro OER values decreased gradually, then rapidly with increasing LET, while in vivo OER values decreased more slowly with increasing LET. A series of works by Heidelberg [63,64,65] involving clamped prostate carcinomas at various differentiation levels in rats provided additional evidence for the decreasing impact of tumor intrinsic characteristics, including hypoxia, on the radioresistance against C-ion radiation. They found that OER was higher in vitro than in vivo, possibly caused by factors in the microenvironment, such as immune response. With regard to the behavior of these factors under hypoxia, knowledge remains very scarce. For example, the only attempt to investigate immunogenic cell death following C-ion irradiation was performed in vitro by measuring calreticulin and programmed cell death ligand 1 (PDL1) expression; no impact of carbon radiation in hypoxia was found [66].

### 3.3. Benefits of CIRT for Hypoxic Tumors: Clinical Evidence

Despite the increasing in vitro and in vivo evidence supporting the potential of high-LET radiation to tackle hypoxia, the clinical evidence that C ions radiation increases tumor control probability (TCP) in hypoxic tumors remains scarce. Positive results have been obtained in two highly hypoxic types of tumor: pancreas adenocarcinoma and uterine cervical cancer. Furthermore, the use of carbon ions is being considered for glioblastomas.

#### 3.3.1. Pancreatic Cancer

Characteristic features of pancreatic cancer include the immunosuppressive tumor microenvironment [67] and significantly higher hypoxia levels than most solid tumors [68,69], contributing to increased invasiveness and therapy resistance. A unique feature of pancreatic tumor histology is the extensive desmoplasia around the tumor [70], which is the formation of dense fibrotic tissue, leading, among consequences, to an increase in interstitial pressure and compression of blood vessels. Furthermore, in a complex relationship, fibrosis and hypoxia amplify each other, reinforcing the metastatic phenotype. Despite decades of research, the five-year survival rate for locally advanced pancreatic cancers (LAPCs) remains low, at 5–10% [71,72]; photon radiotherapy is inefficient due to the high hypoxia levels and proximity to organs at risk (OARs) [72].

Clinical trials in Japan have shown promising outcomes of LAPC treatment with C ions used in combination with chemotherapy (gemcitabine). At NIRS, in a phase I trial, 26 patients receiving doses of 30 Gy (RBE) up to 36.8 Gy (RBE) followed by a resection in 21 patients had a remarkable 5-year overall survival rate of 42% [73]. Another NIRS trial combined CIRT with concurrent gemcitabine, leading to 2-year overall survival rates of 35% in all patients and 48% in the high-dose group with stage III LAPC [74]. Encouraging results have also been reported from Gunma and Chiba with respect to C-ion reirradiation for the treatment of local recurrences [75,76]. A model of chemoradiotherapy for LAPC showed that Japanese results are superior to those reported in all trials conducted with X-rays in Europe and the USA in terms of TCP [77]. In Europe, there are currently two ongoing trials: the PIOPPO Protocol at the National Center for Oncological Hadrontherapy (CNAO) in Pavia [78] studying the effectiveness of C-ion neoadjuvant treatment and the phase II PACK study at Heidelberg Ion Beam Therapy Center (HIT) [79] investigating carbon-ion treatments for LAPC and locally recurrent pancreatic cancers.

#### 3.3.2. Cervical Cancer

Uterine cervical cancer is another example of hypoxic cancer that can be efficiently treated with C ions. Highly hypoxic in both primary and especially recurrent forms [80], these tumors exhibit poor control rates compared to oxygenated cases [81].

The outcomes of the clinical CIRT treatments in 1995–2000 at NIRS in Japan, during which pO_2_ was measured with an oxygen electrode in individual patients, show comparable disease-free survival and local control rates between hypoxic and oxygenated tumors, indicating the reduction in hypoxia-induced tumor radioresistance with C-ions [82]. This was the first clinical attempt to directly demonstrate the ability of LET beams to decrease OER and successfully control hypoxic tumors. More recent findings from Japan’s working group on gynecological tumors [83,84] further demonstrate favorable 5- and 10-year local control rates for cervical carcinomas treated with carbon beams.

#### 3.3.3. Glioblastoma

Glioblastoma (GBM), the most common type of brain tumor [85], remains one of the most resistant cancer types. With the standard of care being surgical resection followed by adjuvant radiotherapy with temozolomide, the median survival time of newly diagnosed patients is only 15 months [86]. While GBM resistance is explained primarily by infiltrative growth in surrounding brain tissue, it is further impacted by significant levels of hypoxia with pO_2_ values below 10 mmHg [87]. In this regard, C-ions are seen as a solution to escalate the GBM dose while minimizing normal brain tissue damage [88,89,90].

A phase I/II clinical trial at NIRS demonstrated the potential of C-ion boosting to increase the median survival times for GBM patients up to 26 months in the high-dose group (24.8 Gy (RBE)) [91]. Later, HIT (Heidelberg) initiated the CLEOPATRA [92] and CINDERELLA [93] trials to compare C-ion boost to proton boost and evaluate the potential of CIRT for recurrent tumors, respectively; the final results are yet to be published. Last but not least, the phase I/III trial at Shanghai Proton and Heavy Ion Center (SPHIC) explored the potential of delivering a C-ion boost prior to the initiation of standard chemoradiotherapy, when hypoxia is at its highest levels [94]. 

## 4. Tumor Reoxygenation and Local Oxygenation Changes

One of the main rationales for fractionation in radiotherapy is tumor reoxygenation [95], i.e., supplying oxygen to the surviving previously hypoxic tumor regions (Figure 2), which, in certain conditions, can outweigh the effects of sublethal injury repair and regeneration.

A review of the outcomes of radiotherapy coupled with hypoxia imaging with PET tracers for different tumor sites (primarily head and neck) shows that in all the analyzed studies tumor hypoxia, was reduced in the treatment course [96]. A significant reduction in hypoxia level often occurs after two weeks of treatment, and the status of reoxygenation at this time point can be indicative of treatment success or failure [97,98,99,100].

Two types of oxygenation have been described [54]: slow reoxygenation of chronically hypoxic cells resulting from tumor shrinkage and changes in acute hypoxia between treatment fractions. The success of carbon treatments for hypoxic tumors can be partly attributed to the careful selection of fractionation schemes, enabling local oxygenation changes to occur between fractions. Therefore, reoxygenation may have contributed to the success of the clinical trials described in Section 3.3.

It is interesting to note that early studies at NIRS showed accelerated reoxygenation after C-ion compared to X-ray irradiation. In particular, a mouse tumor study [101] compared reoxygenation in squamous cell carcinoma and mammary sarcoma in the hind limbs after priming irradiation with either C-ion SOBP or X-rays; the authors observed faster reoxygenation after the carbon treatment for two out of three tested tumors. A similar conclusion was reached in a later study [102], demonstrating that C-ion irradiation reoxygenated fibrosarcomas earlier and deeper in space as compared to X-ray irradiation. An early explanation of these effects was attributed to the so-called random killing of both normoxic and hypoxic cells with carbon ions [103], while a recent study in a rat model explains such effects with increased tumor vessel perfusion and permeability [104].

## 5. Is Carbon LET High Enough?

In treatment planning, several beams of different energies are used to produce an SOBP. Especially for large tumors, this unavoidably leads to a “dilution” of the LET values, which remain high for the single Bragg peak but with a much lower mean value. The highest values of dose-averaged LET (LETd) are typically reached around the planning tumor volume (PTV) edges, remaining at low to medium levels in the core, where hypoxia often occurs, leading to inferior clinical outcomes in larger tumors. Large margins around the gross tumor volume (GTV) leads to a relatively low LET distribution exactly where it is most needed, i.e., in the hypoxic subvolumes of the GTV. Furthermore, equal target dose coverage can be achieved, with plans resulting in very different LET distributions [105]. For C ions, a common range of the LETd in the tumor core is 30–80 keV/µm (Figure 3A–C). According to existing OER(LET, pO_2_) models, this would only slightly affect the OER values. For example, according to the model reported in [52] (Figure 1), for a hypoxia level of 0.5% pO_2_, an LET of 50 keV/µm would only lower the OER from approximately 1.6 to 1.5. Consequently, the effect of oxygen can compromise the treatment outcome not only in X-ray radiotherapy but also in CIRT [54].

A clear correlation between the C-ion LETd distribution and tumor control has been demonstrated in pancreatic cancer [106] (Figure 3D), chondrosarcoma [107] (Figure 3E), and sacral chordoma [108] patients. In those trials, local recurrence was correlated with low LETd values, suggesting that LETd maps can be used to classify patients into groups at high or low risk of recurrences in high-dose regions [109]. However, it must be stressed that low LETd values are associated with large tumors, the prognosis of which is worse than that of small tumors for reasons that have nothing to do with LET maps. In fact, a retrospective analysis of uterine carcinoma patients treated with C ions found no direct correlation between severe rectal complications and LETd values [110].

## 6. Strategies to Maximize Carbon-Ion LET and Their Limitations

### 6.1. Simultaneous Integrated Boost

The only approach currently used in clinical trials that might improve the LETd distribution in C-ion treatments is simultaneously integrated boost (SIB). While the main rationale for this approach based on intensity-modulated radiotherapy (IMRT) with X-rays is to deliver a dose boost to selected target subregions during irradiation, it also shifts the LETd distribution to higher values. In Figure 4, we compare the LETd distributions for a chordoma treatment plan optimized (A) for a uniform PTV dose or (B) for a plan including GTV SIB. As shown by LETd–volume histograms (C), the LETd distribution in the GTV is significantly improved, while almost no changes are visible in the PTV. SIB is already a standard approach in IMRT and is under testing with CIRT in three ongoing clinical trials: for head and neck adenoid cystic carcinoma in at CNAO [111,112] and for prostate [113,114] and pancreatic [115] tumors at SPHIC.

### 6.2. Arc Therapy

Another promising strategy in particle therapy is arc therapy, where beams are delivered from multiple angles using a gantry rotating around the patient. Originally aimed at enhancing dose conformity and plan robustness, arc therapy has the additional benefit of increasing LET values within the tumor [117]. With arc therapy, the maximum LET for C ions moves from the tumor edges to its center, making the plan less sensitive to the presence of hypoxia in the tumor core [56,118].

Arc CIRT requires rotating gantries with high magnetic rigidity compared to proton gantries. Despite the use of novel superconductive technologies, C-ion gantries remain large and expensive. At the moment, only five facilities (one in Germany, two in Japan, and two in South Korea) use C-ion gantries. A potential breakthrough in this field is the possible delivery of radiotherapy to patients in an upright position [119]; however, rotation speed, acceleration, angular range, and the tolerance of the patients to these parameters still need to be addressed [120]. In addition, both gantry and patient rotation need to combine beam raster scanning with the live rotation of the beam or the target.

### 6.3. LET Painting

Other options are available to increase the LETd in the GTV. For example, the so-called blocking technique was proposed at MedAustron for bulky tumors with the idea of introducing field-blocking structures to stop the beams a few centimeters beyond the midplane of the GTV. This approach can increase and concentrate the high LETd in the tumor center while reducing it in adjoining organs at risk, as was demonstrated with pelvic sarcoma plans and is about to be implemented in an upcoming prospective clinical study [121].

For more adaptive and personalized approaches, the detection of hypoxic tumor subtargets and their incorporation into treatment planning are essential, going beyond standard RBE-weighted optimization. “LET painting” is a treatment plan method to adapt particles’ LET throughout the tumor PTV according to oxygenation [105,122]. LET boosting of hypoxic regions is achieved by applying treatment fields with dose ramps in the SOBP, generated following the position and shape of the hypoxic tumor volume identified through functional imaging. This approach, either alone or in combination with dose-painting techniques, has the potential to increase the tumor control probability [123]. NIRS is currently preparing a trial for head and neck tumors aimed at assessing the safety and effectiveness of LET painting, although without coupling the approach with hypoxia imaging [124].

When both pO_2_ and LET are used in the optimization process in combination with modified algorithms to compute the biological effect in a mixed radiation field, we have a “kill painting” [52,53] to achieve uniform tumor cell killing for any oxygenation distribution in the PTV. This optimization requires intratumor oxygenation maps and an OER model describing the corresponding subvolume radiosensitivity, including its LET dependence. The optimizer redistributes particle fluences in different energy slices, effectively increasing LET in the target center.

### 6.4. Multi-Ions

A superior—albeit more intricate—strategy involves the integration of multiple ion beams within a single treatment plan, simultaneously profiting from the physical and radiobiological properties of several relevant ions. In this context, carbon ions can be employed to target a small hypoxic fraction of the tumor or even replaced with heavier ions, such as oxygen ^16^O [125]. A simple approach is to use carbon ions to boost the LET locally in a proton plan [105]. The idea is to increase the LET in the GTV, maintaining low toxicity in normal tissue. An extension of this concept developed at NIRS is intensity-modulated composite particle therapy [126], which optimizes both dose and LET distributions by combining light and heavy ions within a single treatment plan. Followed by the adaptations of the stochastic microdosimetric kinetic used for biological optimization, this approach was further expanded to account for pO_2_-dependence of OER by boosting the LET values in hypofractionated treatments [127]. A similar outcome was achieved with the strategy proposed at GSI, where the abovementioned kill-painting approach was upgraded to handle multiple ion species, with heavy high-LET ions automatically assigned to hypoxic subvolumes and the rest of the tumor dose delivered by the light ion beams [128].

### 6.5. Selective Targeting of Hypoxic Tumor Segments

The treatment of large bulky tumors faces several challenges in both conventional and particle therapy. These tumors tend to be very hypoxic and can sometimes be so large that full dose coverage in radiotherapy is not possible. To tackle such tumors, Slavisa Tubin developed an IMRT approach called PArtial Tumor irradiation targeting HYpoxic segment (PATHY) [129]. The treatment aims to exploit the bystander effect and immune activation in large hypoxic tumors by irradiating the hypoxic subvolume and sparing the peritumoral immune tumor microenvironment. The peritumoral environment contains blood–lymphatic vessels and lymph nodes, the sparing of which would enhance the immune response of the treatment. The first clinical results with IMRT in large, unresectable tumors were very promising [130]. Carbon ions seem to be the ideal modality for PATHY treatment in terms of irradiation precision, reduced OER in the bystander target volume, and stronger immunogenic cell death compared to X-ray irradiation [131]. When he moved to the MedAustron C-ion center, Dr. Tubin and his colleagues started carbon-PATHY studies with the ultimate goal of reducing the tumor volume to the point that surgery is possible [132]. A phase I/II clinical trial is currently ongoing at MedAustron to evaluate the feasibility and effectiveness of Carbon-PATHY for unresectable bulky tumors [133].

A somewhat similar method called microporous radiation was investigated by the researchers at SPHIC. In their in vivo study [134], the authors observed that apart from the growth delay of the primary tumor, microporous radiation limited only to the hypoxic segment of the primary tumor induced robust abscopal effects in the non-irradiated tumor similar to CIRT covering the entire tumor.

### 6.6. Challenges of LET Optimization

Despite the elegance of the solutions described above, their implementation in clinical practice remains hindered by several unresolved questions.

First, most of them require a precise model for the OER dependence on particle LET and tissue oxygenation, and despite the large number of existing models mentioned in Section 3.1, the optimal function remains unknown. Additionally, these models heavily rely on in vitro cell survival data, which might lead to uncertainties in the estimations of in vivo effects. Thus, further efforts are essential to understand the underlying biological, chemical, and physical aspects of hypoxia radioresistance, leading to the development of a more accurate OER model.

Secondly, painting techniques require non-invasive hypoxia imaging tools with high temporal and spatial resolutions. A wide range of techniques has been proposed for this purpose, ranging from more conventional PET- and MRI-based approaches to recently proposed alternatives such as Cherenkov-excited phosphorescence [135] and functional near-infrared spectroscopy [136]. However, the information obtained with any of these methods is still not considered reliable enough for treatment planning [90,137,138]. Among the open issues is that the imaging resolution remains limited with respect to the tumor microenvironment. Differentiation between different forms of hypoxia (diffusion-limited, perfusion-limited, or anemic), which cannot yet be tackled with functional imaging techniques, was proven to be important [139] by employing tumor models with heterogeneous oxygenation [140]. Furthermore, pO_2_ quantification is strongly affected by the choice of a well-oxygenated reference region and its assigned oxygenation level, which is used to normalize tracer uptake using conversion functions [141].

## 7. Comparison to Pharmaceutical Approaches

While CIRT is a physical tool, other approaches are used to overcome tumor hypoxia that need to be mentioned, such as pharmaceutical approaches. A typical example of an OER-reducing drug that can be used in radiotherapy is hypoxic cell-radiation sensitizers nitroimidazoles, which have been found to mimic the effect of oxygen in the radiochemical process and, at a clinically acceptable toxicity level, can theoretically lead to a reduction in OER to values of 1.5–2 [142]. However, the only drug routinely used in clinical practice—and only for treatment of head and neck squamous cell carcinoma in Denmark and Norway—is nitroimidazole-based nimorazole [143]. The meaningfulness of combining nitroimidazoles with CIRT is yet to be evaluated.

Experimental evidence also suggests that targeting HIF-1α together with CIRT could be a promising therapeutic strategy for hypoxic tumors [144]. Notable promising approaches [144] include tumor-targeted lipid-based CRISPR/Cas9 delivery [145] and nanoparticle-based delivery of HIF-1α inhibitors such as lificiguat [146]. Other studies have proposed using inhibitors of tumor heat shock protein 90 (HSP90), which controls the activity and stabilization of HIF-1α [147,148] and has been shown to enhance resistance to CIRT [149]. PU-H71 could be a promising candidate for the role; however, to date, the evidence of its combination with CIRT is limited to in vitro studies in normoxia [149,150]. Another in vitro study involving non-small cell lung cancer cells indicated the increased efficacy of C ions when combined with inhibitors of DNA-dependent protein kinase recruited in the DNA damage response [151].

Hypoxia-activated prodrugs (HAPs) are already in use in clinical trials. HAPs are activated by reduction facilitated by cellular oxidoreductases [152] and can therefore selectively ‘sterilize’ hypoxic tumor cells either by damaging their DNA or targeting proteins associated with tumorigenesis. Based on their design and activation mechanism, HAPs can be divided into several categories, including quinones, nitroaromatics, and aliphatic and heteroaromatic N-oxides (detailed overviews can be found in [153,154]), with tirapazamine, PR-104, and TH-302 among the most prominent. Although combining HAPs with conventional radiation can represent an elegant solution [155], which, in theory, could eliminate the need for CIRT in the treatment of hypoxic tumors, their clinical application has, so far, been hampered by high patient toxicities despite the promising preclinical results. These failures are commonly attributed to the lack of reliable biomarkers predictive of the hypoxia status of the tumor and thus a lack of patient stratification based on the levels of tumor hypoxia [156,157,158,159].

## 8. Conclusions and Future Directions

While the use of protons in radiotherapy is motivated by their physical properties, heavy ions are more justified by their biological characteristics. Cornelius Tobias decided to treat tumors with heavy ions in the LBNL pilot trial, essentially to overcome hypoxia radioresistance [12]. However, the clinical evidence for CIRT’s effectiveness in controlling hypoxic tumors remains limited, and further clinical trials, ideally with quantitative assessment of the levels of hypoxia, are needed to validate its efficacy in a broader range of cancer types.

Furthermore, our knowledge of the OER–LET relationship is largely based on in vitro cell studies. Animal experiments assessing the OER of C-ions show that, despite the generally similar trends, the in vitro and in vivo values and their dependencies on LET differ from each other. This, in turn, raises the need for more in-depth preclinical studies investigating the immune response or cell communication specifically under hypoxic conditions—a topic already clinically exploited in PATHY (see Section 6.5)—as well as at the possible accelerated reoxygenation by C-ions (see Section 4). In such studies hypoxia needs to be induced in natural ways, since the commonly used method of tumor clamping hinders the supply of nutrients and increases the pressure in the tumor capillaries, potentially affecting the measured OER by inducing secondary cell death.

Nevertheless, there is undeniable scientific and clinical evidence of increased particle LET impact on cell killing and tumor response in hypoxia, with multiple approaches proposed to control it. Some proposed approaches, such as SIB, LET blocking, and LET painting, are about to be tested in upcoming clinical trials. The results of these trials will be crucial to decide the next steps. However, if they are not associated with functional hypoxia imaging, e.g., by PET, the results may remain inconclusive. More sophisticated techniques such as carbon-arc and multi-ion therapy require additional improvements in terms of beam delivery and imaging technology.

## Figures and Tables

**Figure 1 cancers-15-04494-f001:**
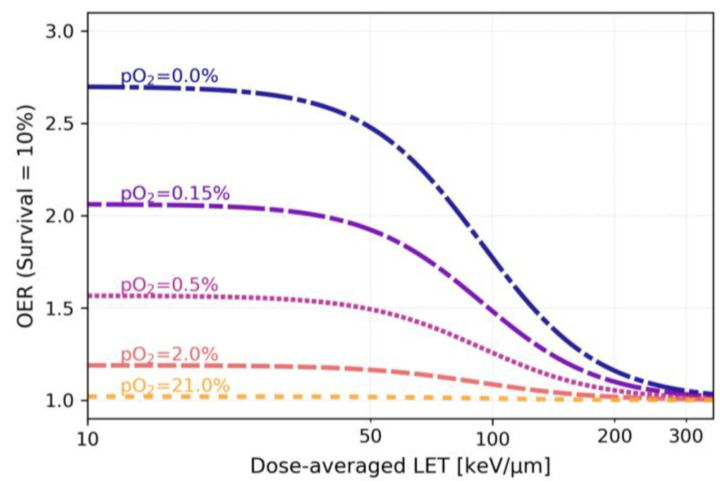
Oxygen enhancement ratio vs. dose-averaged LET at different oxygenation levels (pO_2_) according to the model [52] adapted for the parameters of the CHO cell line.

**Figure 2 cancers-15-04494-f002:**
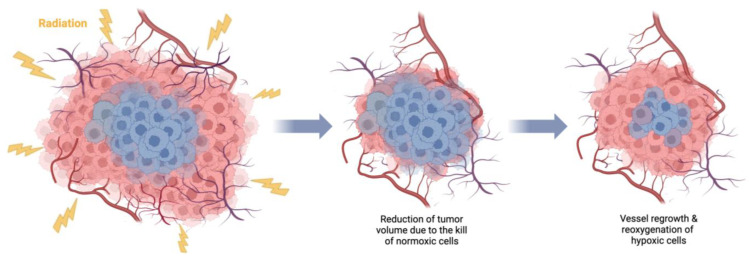
Tumor reoxygenation following irradiation. Death of irradiated normoxic (red color) cells causes tumor shrinkage, which, in turn, leads to the regrowth of blood vessels and supply of previously hypoxic (blue color) cells with oxygen, increasing their radiosensitivity. Created with BioRender.com.

**Figure 3 cancers-15-04494-f003:**
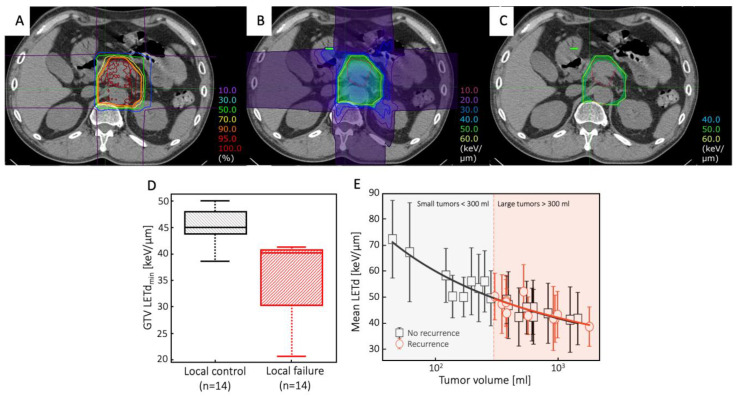
(**A**–**C**) A representative pancreatic tumor treated with carbon-ion radiotherapy: (**A**) dose distribution; (**B**) dLET distribution; (**C**) dLET distribution above 50 keV/μm (adapted from [106]). (**D**) Comparison of minimum values of dLET within the GTVs of pancreatic tumors with local control or local failure [106]. (**E**) average LETd values in the PTVs of chondrosarcomas and their correlation with tumor volumes and recurrences (black squares—non-recurrent cases; red circles—recurrent cases); no recurrencies were observed for tumors smaller than approximately 300 mL [107]. Panels (**A**–**D**) are distributed under Creative Commons CC-BY-NC-ND. Panel E is reproduced with permission of International Institute of Anticancer Research (IIAR journals).

**Figure 4 cancers-15-04494-f004:**
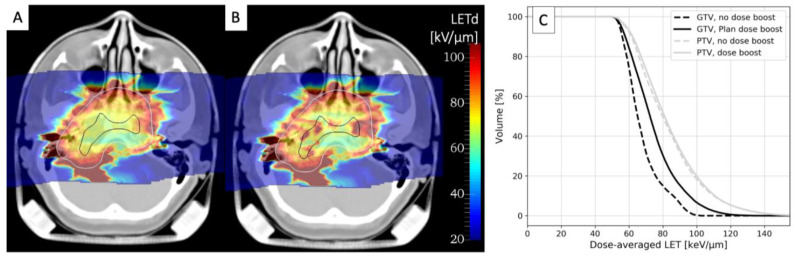
LETd distribution calculated in ^12^C two-field treatment plan for chordoma optimized for either a uniform PTV dose of 3 Gy (RBE) (**A**) or for an additional GTV dose boost of 1.5 Gy (RBE) (**B**). White and black contours represent PTV and GTV, respectively. (**C**) LETd–volume histograms for boosted (solid lines) and non-boosted treatment plans (dashed lines) for tumor GTV (black lines) and PTV (grey lines). Calculations were performed using GSI in-house treatment planning software TRiP98 [116]. Anonymized CT data were retrieved from the pilot project repository of GSI, where treatment was performed, according to German law. Informed consent is waived by the ethical committee of the University of Heidelberg because of the anonymized nature of the research plans.

## Data Availability

Data used for this review were compiled by searching the PubMed and Scopus databases with no limitation on the date of publication. Search terms included “particle therapy” “carbon ions”, “LET”, and “hypoxia”. Particular attention was focused on publications dealing with new research topics. When possible, primary sources are quoted, but review articles that covered the material in greater detail were cited instead.

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
