# Peer review of "Carbon Ions for Hypoxic Tumors: Are We Making the Most of Them?"

_cancers, 2023, doi:10.3390/cancers15184494_

Round 1

Reviewer 1 Report

1. In part 3.3, authors presented the clinical evidence for hypoxia tumors by carbon ion therapy. Pancreatic cancer and cervical cancer were sampled. How about other cancers? As mentioned in ref. 44, hypoxia in glioblastoma cells also impact on carbon ion therapy. The work ( Kong L, Gao J, Hu J, et al. (2019). Carbon ion radiotherapy boost in the treatment of glioblastoma: a randomized phase I/II clinical trial. Cancer Commun (Lond). 39(1):5. doi: 10.1186/s40880-019-0351-2. PMID: 30786916; PMCID: PMC6383247.),  also presented the benefit of carbon ion therapy for hypoxia tumor and excluded in the paper. Please check and list all clinical trials.

 2. In part 6 ‘Strategies to maximize the carbon ion LET and their limitations’, authors listed ‘Simultaneous integrated boost’, ‘Arc therapy’, ‘LET painting’, ‘Multi-ions’, ‘Carbon PATHY’ and ‘Challenges of LET optimization’. How about spatial fractionated carbon ion therapy? Robust abscopal effect can be triggered by carbon-ion microporous radiation in reference (Huang Q, Sun Y, Wang W, Lin L-C, Huang Y, Yang J, Wu X, Kong L and Lu JJ (2020) Biological Guided Carbon-Ion Microporous Radiation to Tumor Hypoxia Area Triggers Robust Abscopal Effects as Open Field Radiation. Front. Oncol. 10:597702. doi: 10.3389/fonc.2020.597702) . Is there any work for biological RBE optimization? Is there benefits from combination technology like multi-ions arc therapy or others?

 3. In part 7 ‘Comparison to pharmaceutical approaches’, some of prodrugs were listed. In reference (Klein, C., Dokic, I., Mairani, A. et al. Overcoming hypoxia-induced tumor radioresistance in non-small cell lung cancer by targeting DNA-dependent protein kinase in combination with carbon ion irradiation. Radiat Oncol 12, 208 (2017). https://doi.org/10.1186/s13014-017-0939-0), inhibitor of DNA PKi, M3814, sensitizes hypoxic cells to carbon ion therapy. It is better to summarized all studies.

Please check the abbreviation. In line 28 page 1, 12C represents carbon ion, and C-ion also means carbon ion in line 40 page2, and what does ‘12C- ions’ mean in line 41 page2? 

Author Response

Reviewer 1

  1. In part 3.3, authors presented the clinical evidence for hypoxia tumors by carbon ion therapy. Pancreatic cancer and cervical cancer were sampled. How about other cancers? As mentioned in ref. 44, hypoxia in glioblastoma cells also impact on carbon ion therapy. The work ( Kong L, Gao J, Hu J, et al. (2019). Carbon ion radiotherapy boost in the treatment of glioblastoma: a randomized phase I/II clinical trial. Cancer Commun (Lond). 39(1):5. doi: 10.1186/s40880-019-0351-2. PMID: 30786916; PMCID: PMC6383247.),  also presented the benefit of carbon ion therapy for hypoxia tumor and excluded in the paper. Please check and list all clinical trials.
  • Indeed, glioblastoma is another example of hypoxic tumor for which C-ions are a promising solution; however, it is not mainly the hypoxia that defines the poor prognosis, but rather the tumor infiltration. This is why we did not include it in the overview in the first place. However, now we have added a new subsection following your suggestions.

  1. In part 6 ‘Strategies to maximize the carbon ion LET and their limitations’, authors listed ‘Simultaneous integrated boost’, ‘Arc therapy’, ‘LET painting’, ‘Multi-ions’, ‘Carbon PATHY’ and ‘Challenges of LET optimization’. How about spatial fractionated carbon ion therapy? Robust abscopal effect can be triggered by carbon-ion microporous radiation in reference (Huang Q, Sun Y, Wang W, Lin L-C, Huang Y, Yang J, Wu X, Kong L and Lu JJ (2020) Biological Guided Carbon-Ion Microporous Radiation to Tumor Hypoxia Area Triggers Robust Abscopal Effects as Open Field Radiation. Front. Oncol. 10:597702. doi: 10.3389/fonc.2020.597702) . Is there any work for biological RBE optimization? Is there benefits from combination technology like multi-ions arc therapy or others?
  • Thank you for pointing out to the article. We have modified the title of the subsection 6.5 and included this approach there. With regards to the RBE optimization, we have mentioned the kill-painting approach that considers both the RBE and the OER of the radiation fields. Multi-ion arc may very well be considered in the future, but at the moment it seems too be premature.
  1. In part 7 ‘Comparison to pharmaceutical approaches’, some of prodrugs were listed. In reference (Klein, C., Dokic, I., Mairani, A. et al. Overcoming hypoxia-induced tumor radioresistance in non-small cell lung cancer by targeting DNA-dependent protein kinase in combination with carbon ion irradiation. Radiat Oncol 12, 208 (2017). https://doi.org/10.1186/s13014-017-0939-0), inhibitor of DNA PKi, M3814, sensitizes hypoxic cells to carbon ion therapy. It is better to summarized all studies.
  • Thank you for pointing out this study, we have included the respective reference in the end of the second paragraph of the section.

Comments on the Quality of English Language

Please check the abbreviation. In line 28 page 1, 12C represents carbon ion, and C-ion also means carbon ion in line 40 page2, and what does ‘12C- ions’ mean in line 41 page2? 

  • We have made the abbreviations more consistent throughout the manuscript.

Reviewer 2 Report

This is a well-written paper on a very interesting and timely topic regarding the use of carbon ion beams to tackle hypoxic tumors. Just a couple of comments that can be addressed in the final version:

- The oxygen track hypothesis (as opposed to the oxygen fixation hypothesis) needs some elaboration since it will be an unknown concept to most readers.

- Some drawbacks of C ions (e.g. fragmentation tail) related to secondary cancer and normal tissue toxicity must be mentioned.

- Figure 3 needs further explanation (not with respect to the LET-dependence but to the pO2 dependence).

Author Response

Reviewer 2

 This is a well-written paper on a very interesting and timely topic regarding the use of carbon ion beams to tackle hypoxic tumors. Just a couple of comments that can be addressed in the final version:

- The oxygen track hypothesis (as opposed to the oxygen fixation hypothesis) needs some elaboration since it will be an unknown concept to most readers.

  • We have added the definition of the oxygen in the track hypothesis and an additional reference to make the last sentence of the paragraph clearer.

- Some drawbacks of C ions (e.g. fragmentation tail) related to secondary cancer and normal tissue toxicity must be mentioned.

  • We have added two sentences regarding the fragmentation and range uncertainties in the introduction.

- Figure 3 needs further explanation (not with respect to the LET-dependence but to the pO2 dependence).

  • We have added an example of the pO2 decrease estimation for the LET range given in the figure 3 A-C.

Reviewer 3 Report

General comments:

The review article by Sokol and Durante is topical, though the information and flow sometimes feel like written in haste. The paper requires an English language revision - some linguistic errors are listed in the Specific comments, though the list is not comprehensive. I also have some suggestions regarding the structure of the paper. Below are my comments:

(1) The authors must specify how did they conduct the collection of the literature as this is not a systematic review. If this work was meant to be a scoping review, please mention it in the Introduction, with a clear goal of the paper.

(2) A more natural flow of the Introduction would consist of a presentation of hypoxia and hypoxia-induced challenges, followed by potential solutions - such as carbon ion therapy (not the other way around).

Something along these lines:

1. The challenge of radiobiological hypoxia

2. Carbon ions in radiotherapy: physical and biological properties

3. CIRT for hypoxic tumors, etc

(3) The title of subsection 3.1 implies that the paragraphs deal with in vitro data only, while there are several references for in silico studies. Add to the title 'in silico' (or modelling) studies.

(4) Section 7 - ' CIRT is a physical tool to overcome hypoxia and must be compared with pharmaceutical approaches' - why MUST be compared? Please explain, as there is no prior mentioning of hypoxia prodrugs / cytotoxins or any other chemical-based agents to tackle hypoxia.

Specific comments:

1.     Line 42 - '...ions in the Bragg peak region' (delete 'at')

2.     Line 45 - '...most of the damage is caused by the free radicals...'

3.     Line 47 - expand on the spread-out Bragg peak as some readers might not be familiar with its meaning.

4.     Line 52 - '...irradiation on the tumor microenvironment...'

5.     Line 68 - '...sharing both the regular and chaotic features of...'

6.     Line 69 - '...it is not able...'

7.     Line 85 - 'Usually, there are two distinguished types of hypoxia: chronic and acute'.

8.     Line 94 - '...role of oxygen in the biological...'

9.     Line 105 - '...is characterised by a sigmoid curve'

10.  Line 116 - '...there is an increased production of the...'

11.  Line 119 - '...calculated by Monte Carlo techniques compare...'

12.  Line 126 - '...have shown many years ago...'

13.  Line 144 - section 3.2 - spell out 'Carbon' in the title. Or be consistent and use CIRT similar to the other section titles.

14.  Line 162 - '..OER in vitro was higher ...'

15.  Line 177 - define desmoplasia

16.  Line 285 - '...it is under testing / investigation...'

17.  Line 287 - in Shanghai? - mention the name of the clinics/institute

18.  Line 366 - correct 'unresectable'

19.  Line 381 - '...to understand the underlying biological, chemical and physical aspects of hypoxia radioresistance...'

20.  Line 394 - '..affected by the choice of a well-oxygenated...'

21.  Line 452 - replace 'unconclusive' with 'inconclusive'.

The paper requires an English language revision.

Author Response

Reviewer 3

The review article by Sokol and Durante is topical, though the information and flow sometimes feel like written in haste. The paper requires an English language revision - some linguistic errors are listed in the Specific comments, though the list is not comprehensive. I also have some suggestions regarding the structure of the paper. Below are my comments:

(1) The authors must specify how did they conduct the collection of the literature as this is not a systematic review. If this work was meant to be a scoping review, please mention it in the Introduction, with a clear goal of the paper.

  • A new section called “review criteria” has been added at the end of the manuscript.

(2) A more natural flow of the Introduction would consist of a presentation of hypoxia and hypoxia-induced challenges, followed by potential solutions - such as carbon ion therapy (not the other way around).

Something along these lines:

  1. The challenge of radiobiological hypoxia
  2. Carbon ions in radiotherapy: physical and biological properties
  3. CIRT for hypoxic tumors, etc
  • Thank you for the suggestion. However, we had a different structure of the introduction in mind, emphasizing the spread and benefits of CIRT, bringing the reader to the problem of hypoxia at the end of the section, and dedicating the following section to the radiobiological details. A change would kind of modify the message we are trying to convey with this paper.

(3) The title of subsection 3.1 implies that the paragraphs deal with in vitro data only, while there are several references for in silico studies. Add to the title 'in silico' (or modelling) studies.

  • Thank you, we have modified the title of the subsection.

(4) Section 7 - ' CIRT is a physical tool to overcome hypoxia and must be compared with pharmaceutical approaches' - why MUST be compared? Please explain, as there is no prior mentioning of hypoxia prodrugs / cytotoxins or any other chemical-based agents to tackle hypoxia.

  • We have rephrased the sentence in the paragraph.

Specific comments:

  1. Line 42 - '...ions in the Bragg peak region' (delete 'at')
  2. Line 45 - '...most of the damage is caused by the free radicals...'
  3. Line 47 - expand on the spread-out Bragg peak as some readers might not be familiar with its meaning.
  4. Line 52 - '...irradiation on the tumor microenvironment...'
  5. Line 68 - '...sharing both the regular and chaotic features of...'
  6. Line 69 - '...it is not able...'
  7. Line 85 - 'Usually, there are two distinguished types of hypoxia: chronic and acute'.
  8. Line 94 - '...role of oxygen in the biological...'
  9. Line 105 - '...is characterised by a sigmoid curve'
  10. Line 116 - '...there is an increased production of the...'
  11. Line 119 - '...calculated by Monte Carlo techniques compare...'
  12. Line 126 - '...have shown many years ago...'
  13. Line 144 - section 3.2 - spell out 'Carbon' in the title. Or be consistent and use CIRT similar to the other section titles.
  14. Line 162 - '..OER in vitro was higher ...'
  15. Line 177 - define desmoplasia
  16. Line 285 - '...it is under testing / investigation...'
  17. Line 287 - in Shanghai? - mention the name of the clinics/institute
  18. Line 366 - correct 'unresectable'
  19. Line 381 - '...to understand the underlying biological, chemical and physical aspects of hypoxia radioresistance...'
  20. Line 394 - '..affected by the choice of a well-oxygenated...'
  21. Line 452 - replace 'unconclusive' with 'inconclusive'.

Comments on the Quality of English Language

The paper requires an English language revision.

  • We have corrected the typos mentioned in the list (green) and performed an additional revision of the manuscript (extra edits are marked with orange color).

Round 2

Reviewer 3 Report

The authors have considered most comments raised by this reviewer. The readability of the article has improved.  

Author Response

We thank the reviewer for his/her positive assessment